Corrected: Publisher correction

# Serotonin signals through a gut-liver axis to regulate hepatic steatosis

Wonsuk Choi[1], Jun Namkung[1,2], Inseon Hwang[3], Hyeongseok Kim[1], Ajin Lim[1], Hye Jung Park[4], Hye Won Lee[4], Kwang-Hyub Han[4], Seongyeol Park[1], Ji-Seon Jeong[5], Geul Bang[6], Young Hwan Kim[6], Vijay K. Yadav[7], Gerard Karsenty [7], Young Seok Ju [1], Chan Choi[8], Jae Myoung Suh[1,3], Jun Yong Park [4], Sangkyu Park[1,9] & Hail Kim [1,3,10]

Nonalcoholic fatty liver disease (NAFLD) is increasing in worldwide prevalence, closely tracking the obesity epidemic, but specific pharmaceutical treatments for NAFLD are lacking. Defining the key molecular pathways underlying the pathogenesis of NAFLD is essential for developing new drugs. Here we demonstrate that inhibition of gut-derived serotonin synthesis ameliorates hepatic steatosis through a reduction in liver serotonin receptor 2A (HTR2A) signaling. Local serotonin concentrations in the portal blood, which can directly travel to and affect the liver, are selectively increased by high-fat diet (HFD) feeding in mice. Both gut-specific Tph1 knockout mice and liver-specific Htr2a knockout mice are resistant to HFD-induced hepatic steatosis, without affecting systemic energy homeostasis. Moreover, selective HTR2A antagonist treatment prevents HFD-induced hepatic steatosis. Thus, the gut TPH1-liver HTR2A axis shows promise as a drug target to ameliorate NAFLD with minimal systemic metabolic effects.

[1] Graduate School of Medical Science and Engineering, KAIST, Daejeon 34141, Republic of Korea. [2] Department of Biochemistry, Yonsei University Wonju College of Medicine, Wonju 26426, Republic of Korea. [3] Biomedical Science and Engineering Interdisciplinary Program, KAIST, Daejeon 34141, Republic of Korea. [4] Department of Internal Medicine, Yonsei University College of Medicine, Yonsei Liver Center, Severance Hospital, Seoul 03722, Republic of Korea. [5] Center for Bioanalysis, Division of Metrology for Quality of Life, Korea Research Institute of Standards and Science, Daejeon 34113, Republic of Korea. [6] Biomedical Omics Group, Korea Basic Science Institute, Chungbuk 28119, Republic of Korea. [7] Department of Genetics and Development, Columbia University Medical Center, New York, NY 10032, USA. [8] Department of Pathology, Chonnam National University Medical School, Gwangju 61469, Republic of Korea. [9] Department of Biochemistry, College of Medicine, Catholic Kwandong University, Gangneung 25601, Republic of Korea. [10] KAIST Institute for the BioCentury, KAIST, Daejeon 34141, Republic of Korea. These authors contributed equally: Wonsuk Choi, Jun Namkung. Correspondence and requests for materials should be addressed to J.Y.P. (email: drpjy@yuhs.ac) or to S.P. (email: 49park@cku.ac.kr) or to H.K. (email: hailkim@kaist.edu)

Nonalcoholic fatty liver disease (NAFLD) refers to a group of conditions characterized by excessive fat accumulation in the liver. It has become a serious public health problem globally, affecting more than half a billion people worldwide[1], and is a leading cause of liver transplantation and hepatocellular carcinoma[2,3]. However, no pharmacological agents have been specifically approved for the treatment of NAFLD[4–7]. Therefore, it is necessary to better understand the mechanisms that contribute to the development of NAFLD.

Serotonin (5-hydroxytryptamine (5-HT)) is a monoamine neurotransmitter that modulates central and peripheral functions. It is synthesized from the essential amino acid tryptophan and its production is regulated by the activity of tryptophan hydroxylase (TPH) and the availability of tryptophan[8–10]. The two distinct isoforms of TPH show mutually exclusive tissue expression patterns: TPH1 in peripheral non-neuronal tissues and TPH2 in neurons of the central and enteric nervous system[11]. Since 5-HT cannot cross the blood brain barrier, central and peripheral 5-HT systems are functionally separated[12]. Most of the peripheral 5-HT is synthesized by TPH1 in enterochromaffin cells of the gut. Once released to the blood circulation, the majority of 5-HT is taken up and sequestered into platelets, whereas the rest of 5-HT enters the systemic circulation and reaches peripheral tissues in free form[13]. Once entering systemic circulation, most of the free 5-HT is metabolized in liver and lung maintaining free 5-HT level in peripheral blood at very low levels[12,14,15]. 5-HT produced by TPH1 in gut and other peripheral tissues has been shown to exert specific biological effects dependent on the context of target tissues, e.g., adipose tissues, pancreatic β cells, bone, and liver[16–21].

While various functions of peripheral 5-HT are known, the direct role of 5-HT in regulating hepatic lipid metabolism in vivo is not well understood. Inhibiting peripheral 5-HT synthesis has been proposed to prevent NAFLD indirectly by protecting mice from obesity via thermogenesis in brown adipose tissue (BAT)[16]. Increased energy expenditure in BAT is thought to reduce hepatic lipid influx and thus prevent the development of NAFLD. In contrast, other studies demonstrated a direct effect of 5-HT on hepatic lipid accumulation in an in vitro model[22]. More recently, we have shown that short-term treatment of TPH inhibitors prevented the development of hepatic steatosis in mice fed a high carbohydrate diet without increasing energy expenditure in adipose tissues[23]. These data prompted us to further explore the detailed mechanism and function of 5-HT signals regarding liver energy metabolism in vivo.

Here we show a novel role for gut-derived serotonin (GDS), through direct actions on liver, in the pathogenesis of hepatic steatosis.

## Results

**GDS regulates HFD-induced hepatic steatosis.** We first examined the expression of genes involved in 5-HT metabolism in order to evaluate whether liver could produce 5-HT. Unlike *Slc6a4*, *Maoa*, and *Maob*, neither *Tph1* nor *Tph2* was expressed in the liver, suggesting that the hepatocyte can transport and metabolize 5-HT but is not likely to produce 5-HT (Fig. 1a). Since nutrient infusion induces 5-HT production in the gut in human and the liver is the first organ to encounter GDS via the portal vein, free 5-HT levels in portal blood may reach concentrations sufficient to have endocrine effects on the liver[24]. To test this notion, we measured 5-HT levels in portal blood and peripheral blood of human subjects. Although the absolute levels of blood 5-HT varied among individuals, 5-HT concentrations were higher in portal blood relative to peripheral blood (Fig. 1b and Supplementary Fig. 1a, b). Also, the ratio of 5-HT between portal and

peripheral blood showed a tendency of positive correlation with markers associated with NAFLD such as blood levels of alanine transferase (ALT), gamma-glutamyltransferase (GTP), triglycerides (TG), and transient elastography controlled attenuation parameters (CAP) (Supplementary Fig. 1c)[25–27]. Next, we assessed changes in 5-HT availability along the gut–liver axis on the liver in mice presenting with hepatic steatosis after 8 weeks of high-fat diet (HFD) feeding. Notably, HFD feeding increased both *Tph1* expression and tissue 5-HT level in the gut, and free 5-HT levels in portal blood (Fig. 1c–e). These data suggest that GDS production and portal 5-HT concentration can contribute to the development of NAFLD in both humans and mice.

In order to investigate the functional role of GDS in hepatic steatosis in vivo, we generated gut-specific *Tph1* knockout (KO) (*Villin-Cre*[+/−]; *Tph1*[flox/flox], herein named *Tph1* GKO) mice (Supplementary Fig. 1d, e) and induced hepatic steatosis with 8 weeks of HFD feeding. *Tph1* GKO mice showed no histological difference in liver compared to wild-type (WT) littermates (*Tph1*[flox/flox]) in standard chow diet (SCD) feeding conditions (Supplementary Fig. 1f). However, when hepatic steatosis was induced with 8 weeks of HFD feeding, hepatic lipid droplet accumulation, NAFLD activity score (NAS), and hepatic TG levels were dramatically reduced in the liver of HFD-fed *Tph1* GKO mice (Fig. 1f–h)[28], indicating that reduced GDS production inhibits the progression of NAFLD.

**GDS does not affect systemic energy metabolism.** Hepatic steatosis develops when the rate of fatty acid (FA) input (influx and synthesis with subsequent esterification to TG) is greater than the rate of FA output (oxidation and outflux (very low-density lipoprotein (VLDL) secretion))[29]. Previous studies reported that inhibition of peripheral 5-HT synthesis prevents obesity and metabolic dysfunction by increasing energy expenditure in BAT and beige adipocytes in inguinal white adipose tissue (iWAT)[16,17]. As such, we thoroughly examined the systemic metabolic phenotype of HFD-fed *Tph1* GKO mice to evaluate whether increased energy expenditure improved hepatic steatosis indirectly by reducing FA influx in the liver. Upon HFD, *Tph1* GKO mice, in comparison to WT littermates, showed no difference in body weight (BW) gain, glucose tolerance, insulin sensitivity, plasma lipid profiles, and adiposity (Fig. 2a–e, and Supplementary Fig. 1g). BAT, iWAT, and epididymal white adipose tissue (eWAT) of *Tph1* GKO mice showed no difference in organ weight and histology compared to WT littermates (Fig. 2f–k). Especially, no obvious histological changes indicating BAT activation and beige adipogenesis were observed in BAT and iWAT of *Tph1* GKO mice. In addition, neither *Ucp1* expression in BAT and iWAT nor energy expenditure was different between *Tph1* GKO mice and WT littermates (Fig. 2l–n). These data indicate that GDS does not affect systemic energy metabolism in this context. Taken together, systemic energy metabolism does not seem to play a role in ameliorating HFD-induced hepatic steatosis in *Tph1* GKO mice.

**GDS regulates lipogenic pathways in the liver.** In order to identify the relevant metabolic pathways that prevent hepatic steatosis in *Tph1* GKO mice, we performed gene expression analyses for an extensive array of metabolic markers. The expression of genes involved in lipogenesis were generally decreased in the liver of HFD-fed *Tph1* GKO mice in comparison to WT littermates (Fig. 3a). However, the expression of genes involved in FA uptake, FA oxidation, and VLDL secretion were not different between HFD-fed *Tph1* GKO and WT littermates (Fig. 3b–d). We next subjected *Tph1* GKO and WT littermates to a methionine–choline-deficient (MCD) diet which induces

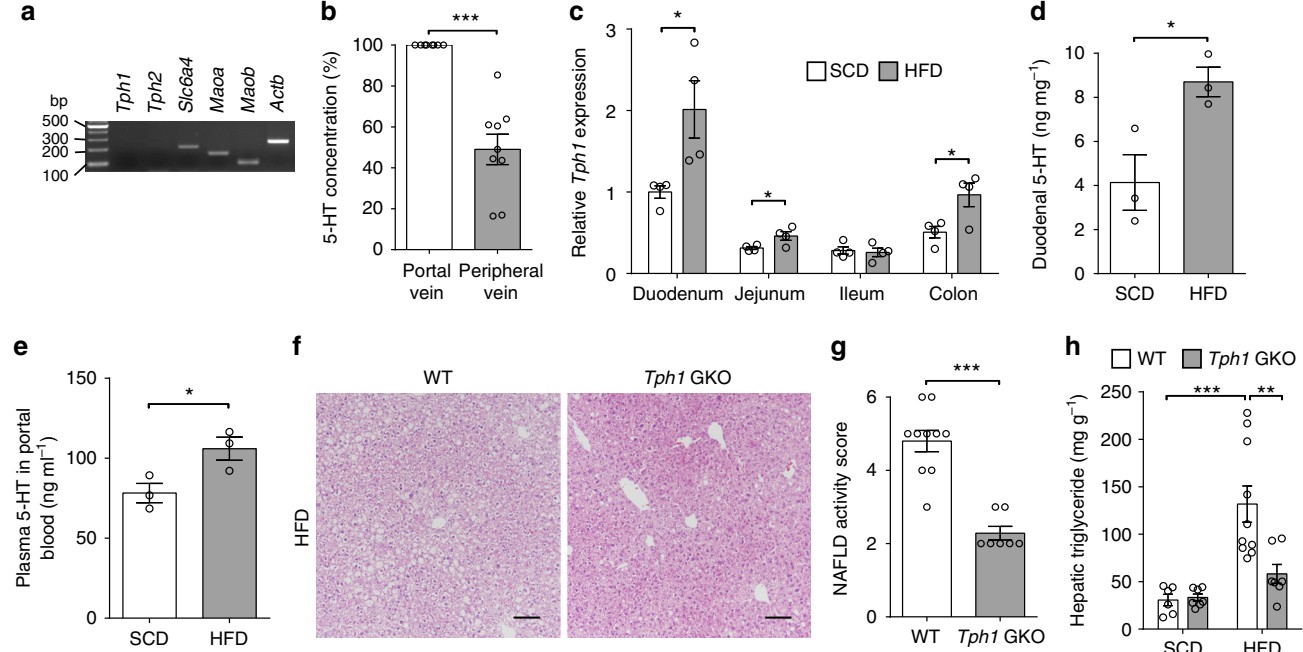

**Fig. 1** Gut-derived 5-HT regulates HFD-induced hepatic steatosis. **a** mRNA expression of genes involved in 5-HT metabolism as assessed by RT-PCR in liver from 8-week-old C57BL/6J mice. **b** Plasma 5-HT concentrations of portal blood and peripheral blood in humans. Portal blood concentration set as 100%; $n = 9$ per group. **c–e** The 12-week-old male C57BL/6J mice were fed standard chow diet (SCD) or high-fat diet (HFD) for 8 weeks. **c** Tph1 mRNA expression in the duodenum, jejunum, ileum, and colon as assessed by qRT-PCR; $n = 4$ per group. **d** Duodenal 5-HT levels; $n = 3$ per group. **e** Plasma 5-HT levels in portal blood; $n = 3$ per group. **f–h** The 12-week-old WT and Tph1 GKO mice were fed SCD or HFD for 8 weeks. **f** Representative liver histology by hematoxylin and eosin (H&E) staining from HFD-fed WT and Tph1 GKO mice. Scale bars, 100 μm. **g** Nonalcoholic fatty liver disease activity score (NAS) of HFD-fed WT and Tph1 GKO mice; $n = 7$–10 per group. **h** Hepatic triglyceride levels; $n = 6$–10 per group. Data are expressed as the means ± SEM. $*P < 0.5$, $**P < 0.01$, $***P < 0.001$, Student's $t$-test (**b–e**, **g**) or one-way ANOVA with post hoc Tukey's test (**h**)

hepatic steatosis by decreasing VLDL secretion rather than increasing FA input in the liver[30]. Liver histology, NAS, and hepatic TG concentrations of Tph1 GKO mice were not different in comparison to WT littermates fed an MCD diet (Fig. 3e–g). As inhibition of GDS synthesis does not prevent hepatic steatosis in Tph1 GKO upon MCD diet, liver FA outflux is unlikely to mediate the GDS effects on liver. Taken together, inhibiting GDS synthesis ameliorates hepatic steatosis by downregulating lipogenic pathways in the liver.

**HTR2A mediates the lipogenic action of GDS in the liver**. The physiological action of 5-HT is mediated through numerous 5-HT receptors (HTRs). To identify the HTR that mediates the role of GDS in the development of hepatic steatosis, we measured the expression of HTR genes in the liver (Supplementary Fig. 2a). Interestingly, among the HTR genes found to be expressed at detectable levels in the liver, only Htr2a expression was markedly increased by HFD feeding (Fig. 4a). To determine whether HTR2A mediates the steatotic effect of GDS in the liver, we generated mice harboring a floxed allele of Htr2a and crossed them with Albumin-Cre mice to generate liver-specific Htr2a KO (Albumin-Cre$^{+/-}$; –Htr2a$^{flox/flox}$, herein named Htr2a LKO) mice (Supplementary Fig. 2b, c). Htr2a LKO mice showed slightly reduced liver size and weight compared with WT littermates (Htr2a$^{flox/flox}$) after 8 weeks of HFD feeding (Fig. 4b, c). Furthermore, HFD-fed Htr2a LKO mice showed decreased hepatic steatosis as examined by histology, NAS, and hepatic TG concentrations (Fig. 4d–f and Supplementary Fig. 2d) without affecting BW, glucose tolerance, insulin sensitivity, and plasma lipid profiles (Supplementary Fig. 2e–h). Similar to Tph1 GKO mice, the expression of genes involved in lipogenesis were

decreased whereas genes involved in FA uptake, FA oxidation, and VLDL secretion were nearly the same in the liver of HFD-fed Htr2a LKO mice compared to WT littermates (Fig. 4g–j). We also subject Htr2a LKO mice to an MCD diet for 6 weeks and found that liver histology, NAS, and hepatic TG levels were not different with MCD diet-fed WT littermates (Fig. 4k–m). These results indicate that Htr2a LKO mice are protected against HFD-induced hepatic steatosis and is a phenocopy of Tph1 GKO mice in terms of hepatic steatosis.

As HTR2B is the most highly expressed HTR in the liver and has been associated with hepatic carbohydrate metabolism[21], we generated liver-specific Htr2b KO (Albumin-Cre$^{+/-}$; Htr2b$^{flox/flox}$, Htr2b LKO) mice and induced hepatic steatosis with 8 weeks of HFD feeding to examine the role of hepatic HTR2B in steatotic effects of GDS in the liver. HFD-fed Htr2b LKO mice showed no difference in liver histology, NAS, and hepatic TG levels compared with WT littermates (Htr2b$^{flox/flox}$) (Supplementary Fig. 3a, b, c). From these results, we confirmed that the protective effect against hepatic steatosis by reducing GDS production was highly specific for hepatic HTR2A.

To further explore the molecular pathways underlying the decreased hepatic lipid accumulation in HFD-fed Htr2a LKO mice, we performed RNA-seq and profiled the liver transcriptomes of HFD-fed Htr2a LKO mice and WT littermates. We analyzed gene ontology (GO) gene sets by gene set enrichment analysis (GSEA). Among 5917 GO gene sets, 3614 gene sets with sufficient number of matched genes were analyzed and 617 gene sets and 27 gene sets were identified to be significantly enriched in WT littermates and Htr2a LKO mice, respectively (Supplementary Data file). Interestingly, gene sets which contribute to inflammation and fibrosis in nonalcoholic steatohepatitis

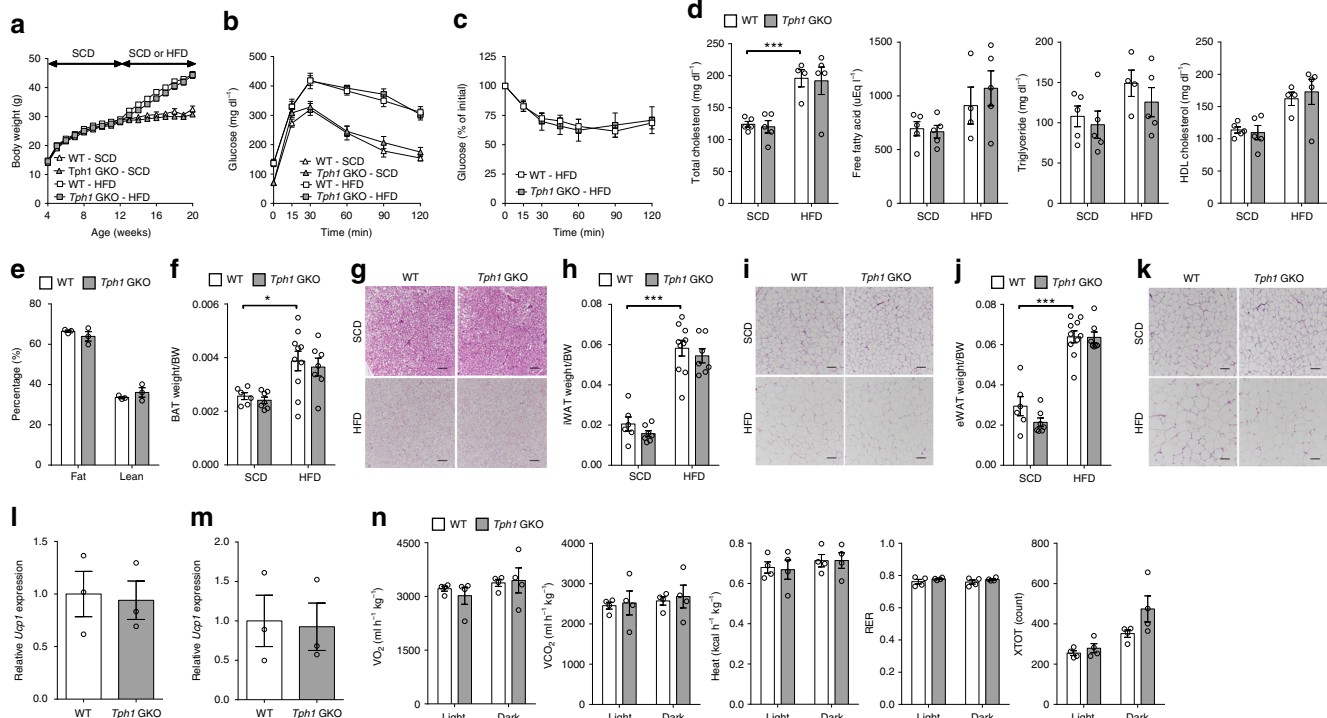

**Fig. 2** Gut-derived 5-HT does not affect systemic energy metabolism. **a–n** The 12-week-old WT and *Tph1* GKO mice were fed SCD or HFD for 8 weeks. **a** Body weight trends; *n* = 6–10 per group. **b** Intraperitoneal glucose tolerance test (IPGTT) after 16 h fasting; *n* = 5 per group. **c** Intraperitoneal insulin tolerance test (IPITT) of HFD-fed WT and *Tph1* GKO mice after 4 h fasting; *n* = 5 per group. **d** Plasma total cholesterol, free fatty acid, triglyceride, and high-density lipoprotein (HDL) cholesterol levels; *n* = 4–5 per group. **e** Percent fat body mass and lean body mass of HFD-fed WT and *Tph1* GKO mice; *n* = 3 per group. **f, h, j** Adipose tissue to body weight ratio of BAT (**f**), iWAT (**h**), and eWAT (**j**); *n* = 6–10 per group. **g, i, k** Representative histology by H&E staining of sections from BAT (**g**), iWAT (**i**), and eWAT (**k**). Scale bars, 100 μm. **l, m** Relative mRNA expression of *Ucp1* assessed by qRT-PCR in BAT (**l**) and iWAT (**m**) of HFD-fed WT or *Tph1* GKO mice; *n* = 3 per group. **n** Metabolic parameters of HFD-fed WT or *Tph1* GKO mice; *n* = 4 per group. Data are expressed as the means ± SEM. *$P < 0.5$, ***$P < 0.001$, Student's *t*-test (**c, e, l, m**) or one-way ANOVA with post hoc Tukey's test (**a, b, d, f, h, j, n**). VO$_2$: oxygen consumption, VCO$_2$: carbon dioxide production, RER: respiratory exchange ratio, XTOT: horizontal motor activity

pathogenesis as well as gene sets that contribute to steatosis were significantly less enriched in *Htr2a* LKO mice (Fig. 5a–c)[31–40]. Quantitative reverse transcription polymerase chain reaction (qRT-PCR) analysis further confirmed that proinflammatory and profibrogenic genes were downregulated in the liver of HFD-fed *Htr2a* LKO mice (Fig. 5d). These results suggest that inhibiting 5-HT signaling through hepatic HTR2A can inhibit the progression of NAFLD.

### HTR2A antagonist ameliorates HFD-induced hepatic steatosis.

Taken together with data from *Tph1* GKO and *Htr2a* LKO mice, we have identified a previously unknown enterohepatic signaling pathway for hepatic steatosis mediated by GDS and hepatic HTR2A. These results led us to evaluate selective HTR2A antagonism as a novel therapeutic strategy for NAFLD. To this end, we tested sarpogrelate which is a selective HTR2A antagonist and mostly used as an antiplatelet agent for the treatment of peripheral arterial disease[41–44]. Mice were fed with HFD for 8 weeks and administered sarpogrelate daily using per os for the same period. As expected, sarpogrelate effectively decreased hepatic TG accumulation (Fig. 6a–c and Supplementary Fig. 4a). In addition, closely resembling the effects of genetic deletion of *Htr2a* in the liver and *Tph1* in the gut, sarpogrelate treatment resulted in decreased expression of genes involved in lipogenesis while genes involved in FA uptake, FA oxidation, and VLDL secretion were not changed in the liver (Fig. 6d–g). We also examined the systemic metabolic effects of sarpogrelate since HTR2A is expressed in tissues outside the liver. Mice treated with

sarpogrelate gained less BW upon HFD and their glucose tolerance was improved, while insulin tolerance was not affected (Supplementary Fig. 4b–d). In addition, adipocyte size in iWAT and crown-like structure in eWAT were decreased in HFD-fed mice treated with sarpogrelate as compared to vehicle-treated HFD-fed mice (Supplementary Fig. 4e, f). These results demonstrate that selective HTR2A antagonist can prevent hepatic steatosis in HFD-fed mice likely through common mechanisms shared with *Tph1* GKO and *Htr2a* LKO mice.

### Discussion

Peripheral 5-HT is emerging as a key regulator of systemic energy metabolism, modulating various physiological roles in multiple metabolic tissues[16–19,21]. Although 5-HT has been known to act on hepatocytes and promote lipid accumulation in vitro[22], controversy still exists whether 5-HT acts directly on liver and regulates hepatic lipid metabolism in vivo. Here we demonstrate that inhibition of GDS synthesis by *Tph1* GKO mice ameliorates hepatic steatosis independent of systemic energy homeostasis and *Htr2a* LKO mice revealed a phenocopy of *Tph1* GKO mice. These results led us to propose GDS as a direct regulator of hepatic lipid metabolism via a gut–liver endocrine axis.

In the present study, we have tested the effects of inhibiting GDS synthesis in two different hepatic steatosis models: HFD which increases fatty acid influx in the liver and MCD diet which reduces fatty acid outflux in the liver. Both *Tph1* GKO and *Htr2a* LKO mice were effective only in HFD-induced liver steatosis without altering systemic energy homeostasis which suggest that

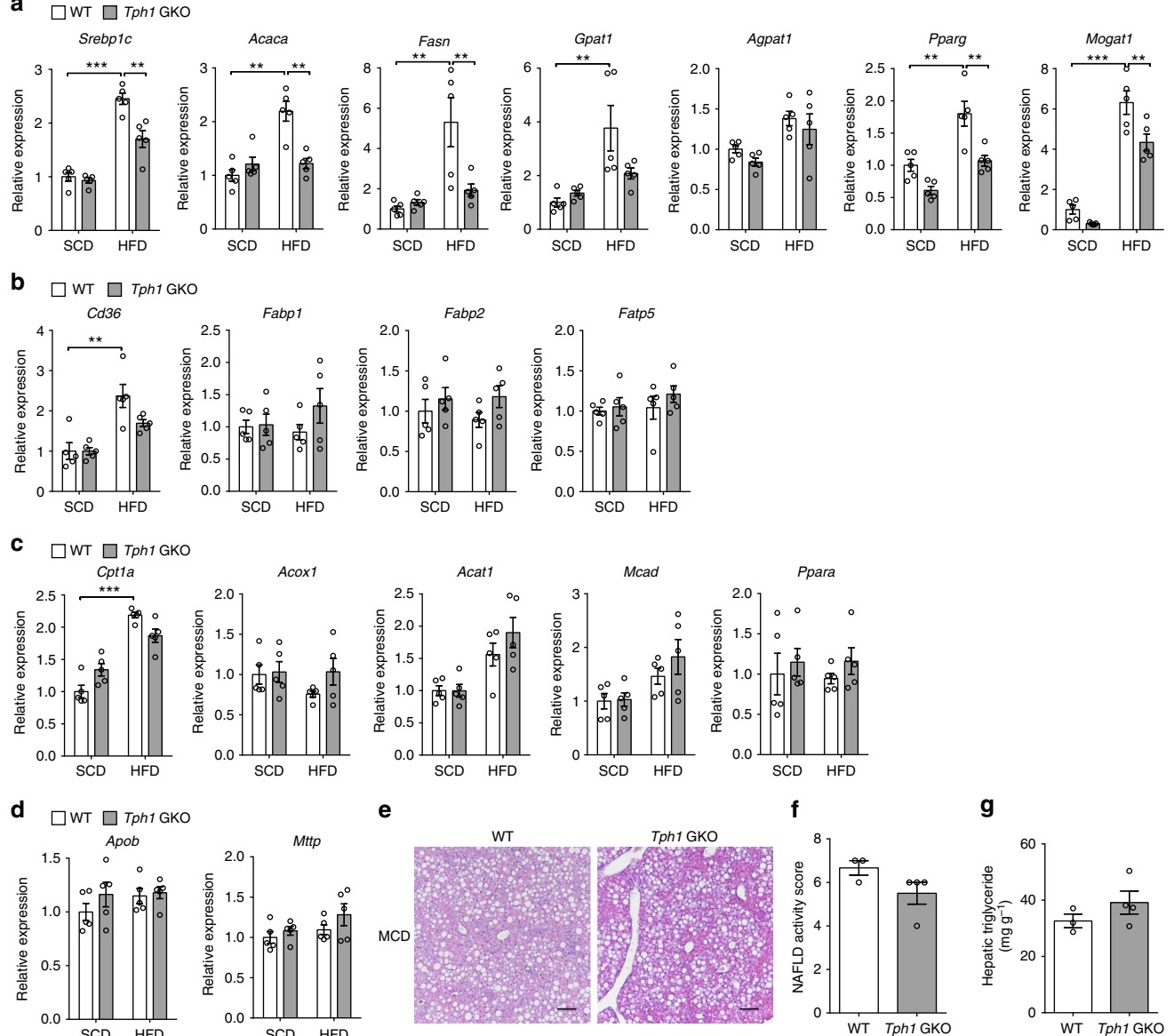

**Fig. 3** Gut-derived 5-HT regulates lipogenic pathways in the liver. **a–d** The 12-week-old WT and *Tph1* GKO mice were fed SCD or HFD for 8 weeks. Relative mRNA expression of genes involved in lipogenesis (**a**), FA uptake (**b**), FA oxidation (**c**), and VLDL secretion (**d**) as assessed by qRT-PCR in liver; n = 5 per group. **e–g** The 8-week-old WT and *Tph1* GKO mice were fed MCD diet for 6 weeks. **e** Representative liver histology by H&E staining. Scale bars, 100 μm. **f** NAS; n = 3–4 per group. **g** Hepatic triglyceride levels; n = 3–4 per group. Data are expressed as the means ± SEM. **P < 0.01, ***P < 0.001, Student's *t*-test (**f**, **g**) or one-way ANOVA with post hoc Tukey's test (**a–d**)

blocking enterohepatic signaling pathways mediated by GDS and HTR2A represses TG accumulation in the liver. It is necessary to further identify detailed molecular mechanisms on how GDS regulates hepatic lipid disposal in the liver and how inhibition of HTR2A signaling reduces hepatic TG accumulation.

Our findings also highlight the functional importance of hepatic HTR2A signaling in the progression of NAFLD. Inactivation of HTR2A signaling in the liver using *Htr2a* LKO attenuated hepatic steatosis independent of systemic energy homeostasis. Moreover, a selective HTR2A antagonist showed efficacy in improving hepatic steatosis as predicted by *Tph1* GKO mice and *Htr2a* LKO mice phenotypes in HFD-induced hepatic steatosis. Thus, hepatic HTR2A is a potential target for therapies aiming to prevent the progression of NAFLD with minimal systemic metabolic effects. In addition, as the mortality increases geometrically with the increasing levels of fibrosis in NAFLD patients and many of genes

involved in proinflammatory and profibrogenic pathways are downregulated in *Htr2a* LKO mice, it would be of interest to test whether HTR2A antagonism can effectively stop the progression of hepatic steatosis to fibrosis in the future[45]. A possible concern regarding the use of HTR2A antagonist for anti-NAFLD treatment is its potential adverse effects on hepatic regeneration after acute liver injury[20]. However, as sarpogrelate shows rare hepatotoxicity in real world practice, this concern might not be a problem to use it as an anti-NAFLD drug.

Crane et al.[16] reported that HFD-fed *Tph1*−/− mice indirectly improved NAFLD through UCP1-dependent thermogenic mechanisms. Oh et al.[17] demonstrated that, similar to *Tph1*−/− mice, inhibition of 5-HT production in adipocytes had an anti-obesity effect by inducing thermogenesis in BAT and iWAT. In the present study, we demonstrated that HFD-fed *Tph1* GKO mice are protected against hepatic steatosis without affecting

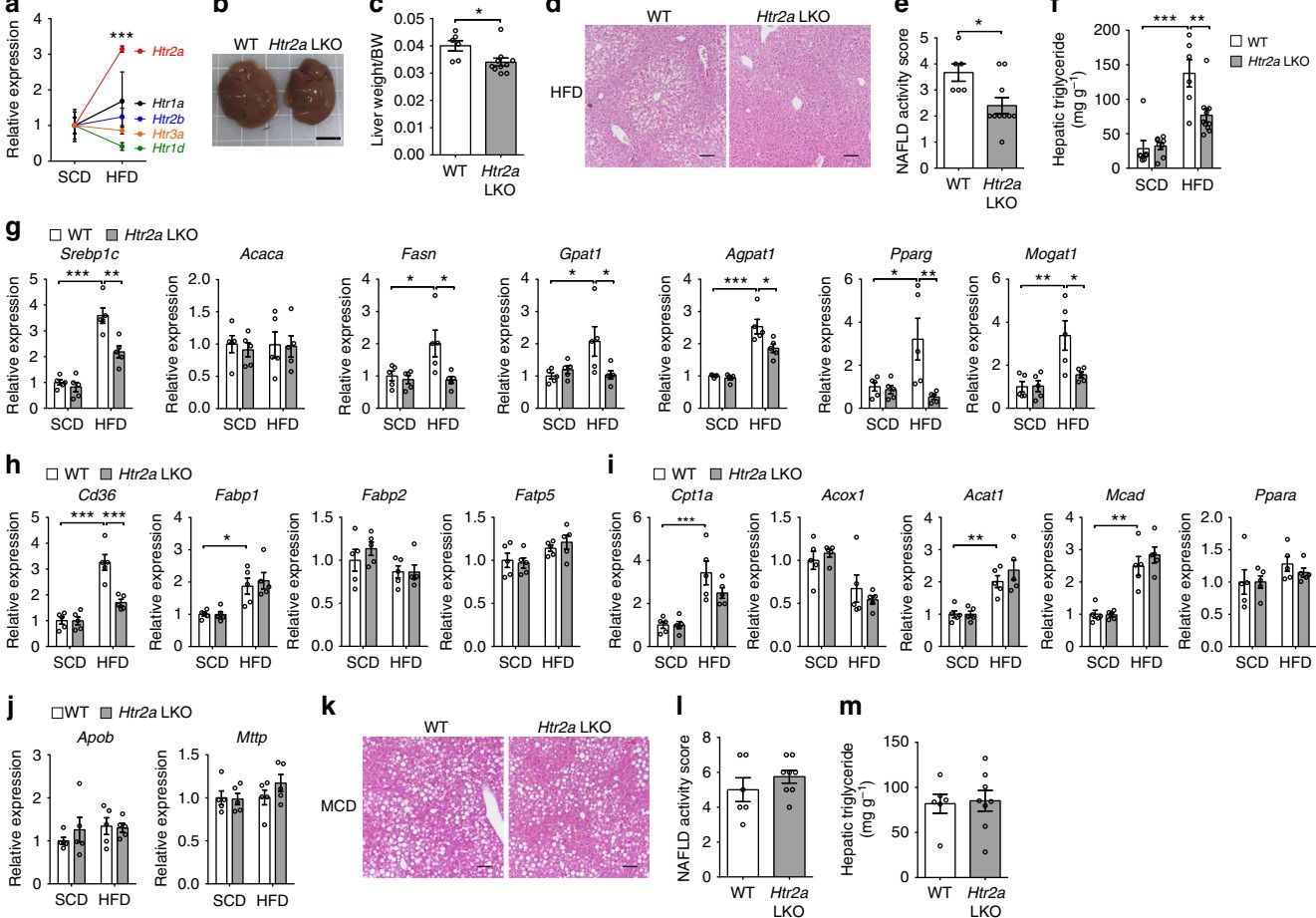

**Fig. 4** HTR2A mediates the lipogenic action of GDS in the liver. **a** Relative mRNA expression of indicated HTRs as assessed by qRT-PCR in liver of SCD- and HFD-fed C57BL/6J mice; $n = 4$ per group. **b–j** The 12-week-old WT and *Htr2a* LKO mice were fed SCD or HFD for 8 weeks. Representative gross liver image (**b**) and liver weight (**c**) of HFD-fed WT and *Htr2a* LKO mice; $n = 6$–10 per group. Scale bar, 1 cm (**b**). **d** Representative liver histology by H&E staining from HFD-fed WT and *Htr2a* LKO mice. Scale bars, 100 μm. **e** NAS of HFD-fed WT and *Htr2a* LKO mice; $n = 6$–10 per group. **f** Hepatic triglyceride levels; $n = 6$–10 per group. **g–j** Relative mRNA expression of genes involved in lipogenesis (**g**), FA uptake (**h**), FA oxidation (**i**), and VLDL secretion (**j**) as assessed by qRT-PCR in liver; $n = 5$ per group. **k–m** The 8-week-old WT and *Htr2a* LKO mice were fed MCD diet for 6 weeks. **k** Representative liver histology by H&E staining. Scale bars, 100 μm. **l** NAS; $n = 6$–8 per group. **m** Hepatic triglyceride levels; $n = 6$–8 per group. Data are expressed as the means ± SEM. *$P < 0.5$, **$P < 0.01$, ***$P < 0.001$, Student's $t$-test (**a**, **c**, **e**, **l**, **m**) or one-way ANOVA with post hoc Tukey's test (**f–j**)

systemic energy homeostasis. Taken together, our current model dictates that the metabolic phenotype of $Tph1^{-/-}$ mice is the sum of the phenotype of *Tph1* GKO mice and adipocyte-specific *Tph1* KO mice. These findings suggest that there may be more various functions of 5-HT in different tissues. Thus, a more detailed study of the different roles of 5-HT using tissue-specific KO strategy will broaden our understanding of the function of this ancient neurotransmitter.

## Methods

**Reagents.** D-Glucose, nonyl phenoxypolyethoxylethanol (NP-40), and sarpogrelate hydrochloride were purchased from Sigma-Aldrich (St Louis, MO, USA). TRIzol reagent was obtained from Invitrogen (Carlsbad, CA, USA).

**Animals and diets.** The *Htr2a*-targeted embryonic stem cell clone was obtained from the European Conditional Mouse Mutagenesis Program (EUCOMM)[46]. The mutant *Htr2a* allele (Htr2a[tm1a(EUCOMM)Hmgu], Mouse Genome Informatics (MGI)[47] ID: 4946779, Supplementary Fig. 2b) contains L1L2_Bact_P cassette located between exons 2 and 3 of the Htr2a gene, which is composed of a flippase recognition target (FRT) site followed by lacZ sequence and loxP site which is followed by human beta-actin promoter-driven neomycin resistance gene, SV40 polyA, a second FRT site, and second loxP site. A third loxP site is inserted downstream of exon 3 of the *Htr2a* gene. Targeted embryonic stem cells were injected into the blastocysts of BALB/c mice. C57BL/6N × BALB/c chimeric

founders carrying the Htr2a[tm1a(EUCOMM)Hmgu] allele were crossed with C57BL/6J mice. The germ line transmissions of the F1 mice were analyzed and the confirmed offspring were crossed with transgenic flippase mice to remove the flanking FRT sites, resulting in *Htr2a*-floxed mice. *Htr2a*-floxed mice were backcrossed and maintained on a C57BL/6J genetic background. Primers for genotyping were as follows: forward primer 5'-TCTCAGACGTGGAAGGGTCT-3' and reverse primer 5'-ACTGGTCACTGCTCAAAGGG-3', which produce products of 252 base pairs (wild-type allele) or 320 bp (*Htr2a*-floxed allele). To generate gut-specific *Tph1* knockout (*Tph1* GKO) mice, Tph1[flox/flox] (MGI: 3837399) mice were crossed with *Villin-Cre* (MGI: 2448639) mice. To generate liver-specific *Htr2a* knockout (*Htr2a* LKO) and liver-specific *Htr2b* knockout (*Htr2b* LKO) mice, *Albumin-Cre* (MGI: 2176228) mice were crossed with Htr2a[flox/flox] mice and Htr2b[flox/flox] (MGI: 3837400) mice. C57BL/6J mice were purchased from the Charles River Japan (Yokohama, Japan). Mice were housed in climate-controlled, specific pathogen-free barrier facilities under a 12 h light–dark cycle, and chow and water were provided ad libitum. Mice were fed SCD (Research Diet (New Brunswick, NJ, USA) D10001), HFD (Research Diet D12492, 60% fat calories), or MCD (Research Diet A02082002BR) diet. All animal experiments were complied with relevant ethical regulations. Experimental protocols for this study were approved by the institutional animal care and use committee at the Korea Advanced Institute of Science and Technology.

**Metabolic analysis.** To measure the metabolic rate, the mice were individually housed in an eight-chamber, open-circuit Oxymax/Comprehensive Lab Animal Monitoring System (Columbus Instruments, Columbus, OH, USA). After 1 day of acclimation, each mouse was assessed for oxygen consumption, carbon dioxide

**Fig. 5** HTR2A mediates the inflammatory and fibrogenic action of GDS in the liver. **a–d** The 12-week-old WT and *Htr2a* LKO mice were fed HFD for 8 weeks. **a–c** GO gene sets were analyzed by GSEA for HFD-fed *Htr2a* LKO mice livers compared with WT littermates livers. Gene sets related to hepatic steatosis (**a**), inflammation (**b**), and fibrosis (**c**); $n = 3$ per group. **d** Relative mRNA expression of genes involved in inflammation and fibrosis as assessed by qRT-PCR in liver; $n = 5$ per group. Data are expressed as the means ± SEM. *$P < 0.5$, **$P < 0.01$, ***$P < 0.001$, one-way ANOVA with post hoc Tukey's test (**d**)

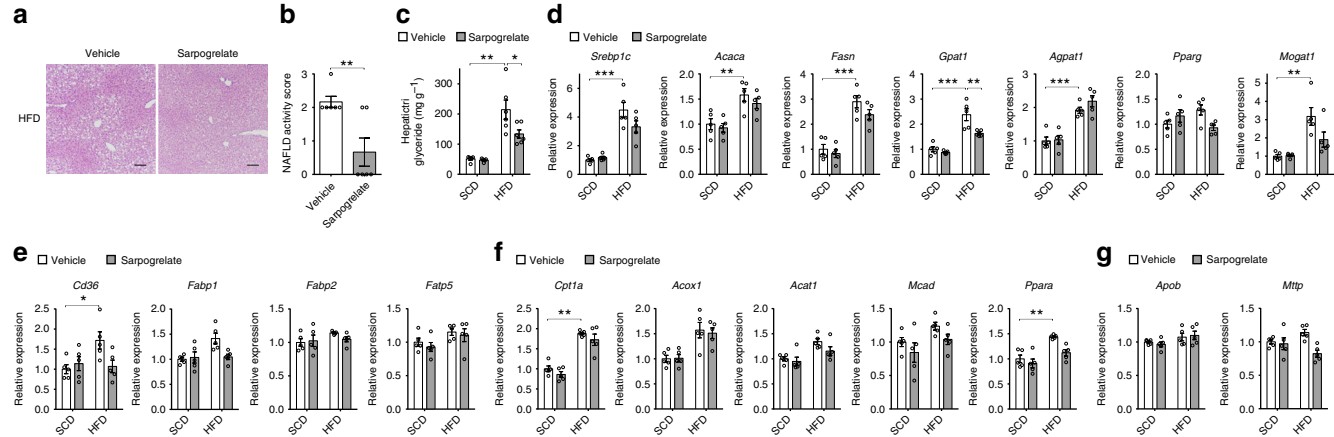

**Fig. 6** HTR2A antagonist ameliorates HFD-induced hepatic steatosis. **a–g** The 12-week-old mice were fed SCD or HFD for 8 weeks and were treated with vehicle or sarpogrelate daily per os. **a** Representative liver histology by H&E staining from HFD-fed vehicle and sarpogrelate treated mice. Scale bars, 100 μm. **b** NAS of HFD-fed vehicle and sarpogrelate-treated mice; $n = 6$ per group. **c** Hepatic triglyceride levels; $n = 5$–6 per group. **d–g** Relative mRNA expression of genes involved in lipogenesis (**d**), FA uptake (**e**), FA oxidation (**f**), and VLDL secretion (**g**) as assessed by qRT-PCR in liver; $n = 5$ per group. Data are expressed as the means ± SEM. *$P < 0.5$, **$P < 0.01$, ***$P < 0.001$, Student's $t$-test (**b**) or one-way ANOVA with post hoc Tukey's test (**c–g**)

production, activity, and food intake for 72 h with access to diet and water ad libitum. Heat production was calculated as $(3.815 + 1.232 \times RER) \times VO_2$, where RER (respiratory exchange ratio) was calculated as $VCO_2 VO_2^{-1}$. Fat mass and lean body mass were measured using a Minispec time-domain nuclear magnetic resonance analyzer (Bruker Optics, Billerica, MA, USA).

**Glucose tolerance test and insulin tolerance test**. For the glucose tolerance test, 2 g kg$^{-1}$ D-glucose in phosphate-buffered saline (PBS) was intraperitoneally injected into overnight-fasted mice. For the insulin tolerance test, 0.75 U kg$^{-1}$ or 1 U kg$^{-1}$ human insulin (Humulin R; Lilly, Indianapolis, IN, USA) was intraperitoneally injected into mice after fasting for 6 h. Blood samples were then obtained

from the tail vein at 0, 15, 30, 45, 60, 90, and 120 min after injection. Glucose concentrations were measured using a Gluco DR Plus glucometer (Allmedicus, Anyang, Korea).

**Blood chemistry analysis**. Blood samples were collected by retro-orbital bleeding into plasma separation tubes with lithium heparin (BD Biosciences, Franklin Lakes, NJ, USA), followed by centrifugation at $1500 \times g$ for 12 min at 4 °C. Enzymatic colorimetric assay kits for total cholesterol (Roche, Basel, Switzerland), high-density lipoprotein cholesterol (Roche), triglyceride (Roche), and free fatty acid (Wako, Osaka, Japan) were used to determine plasma levels on a Cobas 8000 modular analyzer (Roche) at GreenCross LabCell.

**Quantification of mouse serotonin**. To measure mouse platelet poor plasma (PPP) serotonin, blood samples were collected by retro-orbital bleeding or sampling from the portal vein. To acquire a PPP, we centrifuged twice to ensure no platelet contamination of supernatant; initially at $700 \times g$ for 10 min and then $1792 \times g$ for further 10 min. Quantification of serotonin was performed using an enzyme-linked immunosorbent assay kit (IBL International, Hamburg, Germany) according to the manufacturer's instructions. To measure the levels of serotonin in the mouse gut, duodenum tissues were washed in PBS, followed by homogenization in RIPA buffer (Thermo Fisher Scientific, Waltham, MA, USA) using a FastPrep-24 (MP Biomedicals, Santa Ana, CA, USA). The homogenized tissues were centrifuged at $12,000 \times g$ for 5 min at 4 °C. The supernatants were stored at −80 °C until analyses. Serotonin levels in tissue lysates were determined with an enzyme-linked immunosorbent assay kit (IBL International) or liquid chromatography-mass spectrometry method. Tissue serotonin levels were normalized to protein concentrations using a BCA Protein Assay Kit (Thermo Fisher Scientific) of homogenates.

**Quantification of hepatic triglyceride**. Liver tissues were homogenized in 5% NP-40 using FastPrep-24. To solubilize fat, the homogenates were heated to 95 °C for 5 min and cooled at 23 °C, and repeated. Triglyceride Reagent (Sigma-Aldrich) or PBS was added and incubated at 37 °C for 30 min to hydrolyze TG into glycerol. For the colorimetric assay of hydrolyzed TG levels, samples were incubated with Free Glycerol Reagent (Sigma-Aldrich) at 37 °C for 5 min. Differences in absorbance at 540 nm between hydrolyzed or non-hydrolyzed TG were quantified using a glycerol standard (Sigma-Aldrich). TG contents were normalized by the protein concentrations of homogenates, which were measured with a BCA Protein Assay Kit.

**Quantitative RT-PCR analysis**. Total RNA extractions from harvested tissues were performed using TRIzol according to the manufacturer's protocol. After TURBO DNase (Invitrogen) treatment, 2 μg of total RNA was used to generate complementary DNA with Superscript III reverse transcriptase (Invitrogen). Quantitative RT-PCR was performed with Fast SYBR Green Master Mix (Applied Biosystems, Foster City, CA, USA) and a Viia 7 Real-time PCR System (Applied Biosystems) according to the manufacturer's instructions. Gene expression was relatively quantified based on the delta delta Ct (threshold cycle) method with the beta-actin gene as a reference gene. The sequences of primers are given in Supplementary Table 1.

**Histological analysis**. Mouse liver, inguinal white adipose tissue, epididymal white adipose tissue, and interscapular brown adipose tissue were harvested and fixed in 10% neutral buffered formalin solution (Sigma-Aldrich) and embedded in paraffin. The 5 μm-thick tissues sections were deparaffinized, rehydrated, and stained with hematoxylin and eosin (H&E). For quantitative evaluation of hepatic steatosis, NAS, which consists of steatosis, lobular inflammation, and ballooning, was used. NAS was determined by a single certified pathologist, blinded for other information.

**Quantification of human blood serotonin**. All experiments using human participant blood samples were complied with relevant ethical regulations. After previous approval by the institutional review board of Severance Hospital (4-2015-0184), and written, informed consent by all subjects, 9 living donors for liver transplantation were included in this study. Clinical data are given in Supplementary table 2. Body weight and height were measured using a digital scale, and body mass index was calculated by dividing weight (kg) by the square of height (m$^2$). Laboratory parameters including complete blood count and differential count, calcium, phosphorus, glucose, blood urea nitrogen, creatinine (Cr), uric acid, cholesterol, total protein, albumin, alkaline phosphatase, aspartate transferase, ALT, total bilirubin, gamma-GTP, TG, high-density lipoprotein (HDL) cholesterol, low-density lipoprotein (LDL) cholesterol, sodium, potassium, chloride, prothrombin time, and activated prothrombin time were measured within 4 weeks before living donor hepatectomy. Transient elastography was performed using the liver Fibroscan (Echosens, Paris, France). Donor hepatectomy was performed according to standardized procedures. Blood samples for serotonin measurements were drawn simultaneously from the portal vein and peripheral veins during operation. Human PPP serotonin levels were measured using ClinRep high-performance liquid chromatography kit (Recipe, Munich, Germany) at GreenCross LabCell (Yongin, Korea).

**RNA-sequencing**. Total RNA was isolated using a RNeasy Plus Mini kit (QIA-GEN) according to the manufacturer's instructions. The integrity of the total RNA was assessed using an Agilent 2100 Bioanalyzer System (Agilent Technologies) and an Agilent RNA 6000 Nano Kit (Agilent Technologies). Samples with an RNA Integrity Number (RIN) value >8 were selected for use. Libraries were constructed using 1 μg total RNA. The RNA sequencing library was prepared using a TruSeq RNA Sample Prep kit (Illumina) and sequencing was performed using a Illumina NextSeq500 to generate 100 bp paired-end reads. FastQC (FastQC v0.11.3) and cut-adapt (v1.1) were used to filter out sequencing reads of low quality, and the retained reads were mapped to mouse genome build mm10 using TopHat (TopHat v2.0.11) with default parameters. HTseq (v0.9.1) was used to generate raw read counts. GSEA was performed using the GenePattern tool from the Broad Institute, with version of 5.2 of the Molecular Signature Database (MSigDB; http://www.broadinstitute.org/gsea/msigdb/index.jsp). HFD-fed *Htr2a* LKO mice ($n = 3$) were compared with WT littermates ($n = 3$), and representative significantly less enriched gene sets with nominal $P$ value of less than 0.05 are presented.

**Statistics**. All values are expressed as the mean ± standard error of mean and analyses were performed using the SPSS Statistics version 23 (SPSS, Inc., Chicago, IL, USA). Two-tailed Student's $t$-test or one-way analysis of variance (ANOVA) with post hoc Tukey's test were used to compare groups. $P$ values below 0.05 were considered statistically significant. The levels of significance indicated in the graphs are $*P < 0.05$, $**P < 0.01$, and $***P < 0.001$.

**Reporting Summary**. Further information on research design is available in the Nature Research Reporting Summary linked to this article.

## Data availability

The data that support the findings of this study are available from the authors on reasonable request. A Reporting Summary for this Article is available as a Supplementary Information file. Raw data for the RNA sequencing analyses have been deposited in the NCBI's Gene Expression Omnibus (GEO) database (GSE 120662).

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

## Acknowledgements

We thank Drs. Taehee Jo and Byung-Hyun Park for their technical support. We also thank all members of iMOD (Integrated lab of Metabolism, Obesity, and Diabetes) for their helpful discussions. This work was supported by grants from National Research Foundation funded by the Ministry of Science and ICT, Republic of Korea (grant numbers: NRF-2013M3A9D5072550, NRF-2014M3A9D8034464, NRF-2015M3A9B3028218 to Hail Kim, NRF-2016M3A9B6902871 to Hail Kim and J.M.S., NRF-2017R1A5A2015369 to J.N., and NRF-2018R1A2B6001742 to Sangkyu Park), Korea Health Industry Development Institute funded by the Ministry of Health & Welfare, Republic of Korea (grant number: HI15C2859 to J.Y.P.), and KAIST Institute for BioCentury (grant number: N10180027 to Hail Kim).

## Author contributions

W.C., J.N., J.Y.P., S.P., and Hail Kim generated hypothesis, designed the experiments, and wrote the manuscript. W.C., J.N., I.H., Hyeongseok Kim, A.L., J.S.J., G.B., Y.H.K., V.Y., G.K., C.C., J.M.S., Sangkyu Park, and Hail Kim performed and analyzed the animal experiments. W.C., J.N., Seongyeol Park, Y.S.J., and Hail Kim analyzed the RNA-sequencing data. W.C., J.N., H.J.P., H.W.L., K.-H.H., J.Y.P., and Hail Kim analyzed the human data.

## Additional information

**Competing interests:** The authors declare no competing interests.



