## [Peer Review File · Nature Communications]

Reviewers' comments:

Reviewer #1 (Remarks to the Author):

Although the functions of central nervous system serotonin (5-HT) have tended to dominate the research focus, there is now a renewed emphasis on peripheral serotonin as a therapeutic target. This includes the role of serotonin in disorders such as diabetes and obesity as well as in disorders of the gastrointestinal tract, the location where the majority of mammalian 5-HT is produced. In the current study, the authors focus on the role of 5-HT in Nonalcoholic fatty liver disease (NAFLD). Using high-fat diet feeding in mice in combination with a number of experimental approaches (gut-specific Tph1 knockout mice and liver-specific Htr2a knockout mice, selective serotonin receptor 2A [5-HTR2A] antagonist treatment), the study convincingly demonstrates a role for gut derived serotonin synthesis, via liver 5-HT2RA signalling, in hepatic steatosis.

There is a lot to like about this interesting and topical study. The authors have taken a very comprehensive approach to verify their hypothesis with the data from both knockout animals and pharmacological studies supporting their conclusions. I have the following queries and recommendations.

(1) The initial characterisation of the effect of the high fact diet on the liver serotonergic system relies almost exclusively on hepatic gene expression analysis by qRT-PCR. This would benefit from verification at the protein level.

(2) Although the findings are very interesting conceptually, from a practical and translational perspective there may be an issue with therapeutic targeting of hepatic 5-HT2RA since that receptor also has a role in liver regeneration. This point requires inclusion in a revised discussion.

(3) In the introduction, the authors state that 5-HT has exclusive tissue expression patterns, and is produced by TPH1 in peripheral non-neuronal tissues and TPH2 in central neuronal tissues. However, enteric neurons also produce 5-HT and this should also be noted.

(4) The authors indicate that neither Tph1 nor Tph2 were expressed in the liver suggesting that the hepatocyte is not likely to produce 5-HT. This contradicts previous reports of hepatic 5-HT production including evidence of tph gene and protein expression as well as enzyme activity (Valdés-Fuentes et al 2015 <https://physoc.onlinelibrary.wiley.com/doi/abs/10.14814/phy2.12389>).

(5) The study includes measures of 5-HT levels in portal blood and peripheral blood of both mice and human subjects. Can the authors provide more information on the sample processing for this plasma 5-HT analysis since the use of platelet poor plasma, generated by higher than normal centrifugation, is usually required for such studies.

(6) In fig 1b, portal blood 5-HT levels are set at 100%. Why are human portal blood serotonin levels expressed as % when the rodent ones expressed as absolute levels? For consistency, please provide absolute levels of 5-HT in all figures.

(7) There is a lot of variation in the group n numbers for the various measures (e.g. 3 or 4 for the 5-HT measurements in figure 1 but 7-10 for the hepatic tricycleride levels). Why are 5-HT levels not provided for all animals?

Reviewer #2 (Remarks to the Author):

The authors have recently shown that short term treatment of TPH inhibitors prevents development of hepatic steatosis in mice fed a high carbohydrate diet. In this manuscript, the authors extend these studies and explore the mechanism and function of 5-HT signals regarding liver energy metabolism in vivo. Results show link gut-derived 5-HT, through direct actions on liver, to the pathogenesis of hepatic steatosis. In order to investigate the functional role for gut-derived 5-HT the authors generated gut-specific Tph1 knockout (KO) mice and induced hepatic steatosis with 8 weeks of HFD diet. Hepatic lipid droplet accumulation, NAFLD activity score (NAS), and hepatic TG levels were dramatically reduced in the liver of HFD-fed Tph1 GKO mice. In order to identify the relevant metabolic pathways that prevent hepatic steatosis in Tph1 GKO mice, the authors performed gene expression analyses for an extensive array of metabolic markers. In addition, hepatic Htr2a loss-of-function mice were protected against HFD-induced hepatic steatosis. Therefore, the authors evaluated selective HTR2A antagonism as a novel therapeutic strategy for NAFLD. Results presented indicate that selective HTR2A antagonist may prevent hepatic steatosis in HFD-fed mice likely through common mechanisms shared with Tph1 GKO and Htr2a LKO mice. Results highlight the functional importance of hepatic HTR2A signalling in the progression of NAFLD. Inactivation of HTR2A signalling in the liver using Htr2a LKO, attenuated hepatic steatosis independent of systemic energy homeostasis.

Fig. 1:

- 1) Serotonin genes are expressed in the small intestine.
- 2) Serotonin levels are higher in portal as compared to peripheral venous blood.
- 3) High-fat diet increases 1) and 2), as expected
- 4) Tph1G KO results in improved NAS and hepatic triglyceride levels. The authors should include the data showing (as indicated in the text) that circulating serotonin is reduced. This seems to be an important finding.

Fig. 2

- 5) Tph1G KO does not affect systemic energy homeostasis.

Fig. 3

- 6) Tph1 GKO in general express less genes involved in hepatic lipogenesis. This seems to be expected if HTR2A signalling is reduced.

Fig. 4

- 7) Confirms the role of HTR2A.

Fig. 5

Sarpogrelate acts as expected as a HTR2A agonist. This compound has been tested vs. aspirin in secondary prevention of cerebral infarction. A published subgroup analysis did not find any differences in individuals without T2DM, dyslipidaemia which indirectly would have supported the role of HTR2A agonists. Verification of the results in humans is therefore necessary.

Additional comments:

1. The data presented seems rather preliminary and it is necessary to further identify detailed molecular mechanisms on how GDS regulates hepatic lipid disposal in the liver and how inhibition of HTR2A signalling reduces hepatic TG accumulation.
2. As the authors discuss, there may be more various functions of 5-HT in different tissues. Therefore, a more detailed study of the different roles of 5-HT using tissue specific KO strategy is necessary to further our understanding of the function of the neurotransmitter.
3. Do the mice display differences in eating and behaviour? Studies in metabolic cages are necessary.
4. Hepatic lipids should be characterized using lipidomics technology

5. Suppl figure 1 c; significance seems to be driven by one single data point

Reviewer #3 (Remarks to the Author):

Reviewer's comment:

This paper by Choi et al presents studies identifying a role for gut-derived serotonin synthesis in de novo lipogenesis and hepatic steatosis via reduction in liver serotonin receptor 2A (HTR2A) signalling. They showed increased serotonin concentrations in the portal blood of six liver living donor subjects as well as in mice on a high-fat diet. Both gut-specific Tph1 knockout mice and liver-specific Htr2a knockout mice were resistant to short-term HFD-induced hepatic steatosis without having any effects on systemic lipid, glucose, and energy metabolism. Finally, the use of a selective HTR2A antagonist treatment mimicked the protective effects found on the KO on HFD-induced hepatic steatosis. They concluded that gut TPH1-liver HTR2A axis is a promising novel drug target for NAFLD. The manuscript is interesting and of potential significance. There are number of concerns with the current dataset.

Major issues:

1. The human data presented in figures 1 is very hard to interpret. These are living donors were the presence of steatosis/NAFLD is strictly ruled out by extensive work up including liver biopsy that portal blood was obtained from at the time of transplantation. The authors showed correlation with serotonin levels in portal blood with various markers of steatosis and NAFLD progression??
2. In both humans with NAFLD and mice on HFD the main mechanism of steatosis is increased uptake of free fatty acids from circulation mainly via increased lipolysis in the context of systemic and adipose tissue insulin resistance. The current study showed that gut-specific Tph1 knockout mice and liver-specific Htr2a knockout mice on a short-term HFD for eight weeks have no effects on systemic lipid, glucose, and energy metabolism. These results strongly suggest that the effects on steatosis noted are likely to be transient and disappeared with more prolonged feeding. In fact, these diets are typically given 16 to 24 weeks in NAFLD related research. Thus at least a later time point (16 or 24 wks) should be included to better understand the impact of GDS on NAFLD.
3. The suggested effects on the novo lipogenesis in the liver are not fully supported with only limited data on gene expression of various enzymes involved in lipid metabolism provided
4. The studies using sarpogrelate as a therapeutic strategy for NAFLD does not take into account the effects on platelet aggregation and on soluble adhesion molecules induce by this drug that may have a significant impact on the liver phenotype observed. Studies using RNA-based therapy to selectively disrupt the GDS would be warranted.

Response to Reviewers

We thank the reviewers for their careful review of our manuscript. We found the comments and suggestions very helpful. We have tried to comply with all of the comments of the reviewers and highlighted in red color for changes in the revised manuscript. Overall, reviewers raised concerns on the lack of mechanistic insight in the liver and the specificity of 5-HT signaling through gut-liver axis. To address these concerns, we performed RNA-seq analysis in the liver of *Htr2a* LKO mice and analyzed liver specific *Htr2b* KO mice. We also responded to the reviewer's other concerns and outlined those changes below.

Reviewer #1

*Although the functions of central nervous system serotonin (5-HT) have tended to dominate the research focus, there is now a renewed emphasis on peripheral serotonin as a therapeutic target. This includes the role of serotonin in disorders such as diabetes and obesity as well as in disorders of the gastrointestinal tract, the location where the majority of mammalian 5-HT is produced. In the current study, the authors focus on the role of 5-HT in Nonalcoholic fatty liver disease (NAFLD). Using high-fat diet feeding in mice in combination with a number of experimental approaches (gut-specific *Tph1* knockout mice and liver-specific *Htr2a* knockout mice, selective serotonin receptor 2A [5-HTR2A] antagonist treatment), the study convincingly demonstrates a role for gut derived serotonin synthesis, via liver 5-HT₂RA signalling, in hepatic steatosis.*

There is a lot to like about this interesting and topical study. The authors have taken a very comprehensive approach to verify their hypothesis with the data from both knockout animals and pharmacological studies supporting their conclusions. I have the following queries and recommendations.

Comment 1) The initial characterisation of the effect of the high fat diet on the liver serotonergic system relies almost exclusively on hepatic gene expression analysis by qRT-PCR. This would benefit from verification at the protein level

We agree that verification at protein level is beneficial. However, it is difficult to detect 5-HT receptors (HTRs) at protein level because their expression levels are generally very low, usually less than 1 RPKM in RNA-seq. In addition, we haven't had a good antibody for HTR. HTRs are structurally similar with each other and they all have 7 transmembrane domains, which make it difficult to make specific antibody for each HTR. We have been trying to detect HTR by western blot or immunostaining for more than 10 years without any success. This is why lots of papers could not show HTR expression by western blot. Although we did not show HTR expression at protein level, hepatic *Htr2a* expression was well documented by us and others (Cell Metabolism 16, 588-600, 2012). We also showed the disappearance of *Htr2a* expression in *Htr2a* LKO mice along with clear phenotype.

Comment 2) Although the findings are very interesting conceptually, from a practical and translational perspective there may be an issue with therapeutic targeting of hepatic 5-HT_{2A} since that receptor also has a role in liver regeneration. This point requires inclusion in a revised discussion.

We thank reviewer's suggestion. 5-HT is known to play an important role in liver regeneration after acute liver injury, such as hepatectomy, through hepatic HTR_{2A} and HTR_{2B} (Science 312, 104-107, 2006). Thus, HTR_{2A} antagonists need to be carefully used in case patients have acute or chronic liver injury.

As reviewer's suggestion, we revised DISCUSSION as follows (Page 11, Line 4-7): A possible concern regarding the use of sarpogrelate for anti-NAFLD treatment is its potential adverse effects on hepatic regeneration after acute liver injury. However, as sarpogrelate shows rare hepatotoxicity in real world practice, this concern might not be a problem to use it as an anti-NAFLD drug.

Comment 3) In the introduction, the authors state that 5-HT has exclusive tissue expression patterns, and is produced by TPH1 in peripheral non-neuronal tissues and TPH2 in central neuronal tissues. However, enteric neurons also produce 5-HT and this should also be noted.

Yes, enteric neurons express TPH₂ and produce 5-HT.

We revised in manuscript as follows (Page 3, Line 24 ~ Page 4, Line 1-2): The two distinct isoforms of TPH show mutually exclusive tissue expression patterns, TPH₁ in peripheral non-neuronal tissues and TPH₂ in neurons of the central and enteric nervous system.

Comment 4) The authors indicate that neither Tph1 nor Tph2 were expressed in the liver suggesting that the hepatocyte is not likely to produce 5-HT. This contradicts previous reports of hepatic 5-HT production including evidence of tph gene and protein expression as well as enzyme activity (Valdés-Fuentes et al 2015).

As the reviewer indicated, Valdés-Fuentes et al showed that *Tph1* is expressed in rat liver (Physiol Rep 3, e12389). However, we could not detect *Tph1* expression in the liver of C57BL6/J mice. Izikki et. al. also reported that *Tph1* is not expressed in mouse liver (Am J Physiol Lung Cell Mol Physiol 293, L1045-105, 2007). Furthermore, NCBI gene expression database also shows no expression of *Tph1* in the mouse and human liver. We agree that this is somewhat controversial but it could be due to the difference between species.

Comment 5) The study includes measures of 5-HT levels in portal blood and peripheral blood of both mice and human subjects. Can the authors provide

more information on the sample processing for this plasma 5-HT analysis since the use of platelet poor plasma, generated by higher than normal centrifugation, is usually required for such studies.

We measured 5-HT levels in platelet poor plasma (PPP) both in mice and human subjects. To obtain PPP, we centrifuged twice to ensure no platelet contamination of supernatant; initially at 2500 rpm for 10 minutes and then at 4000 rpm for a further 10 minutes.

We revised METHODS as follows (Page 14, Line 6-9): To measure mouse platelet poor plasma (PPP) serotonin, blood samples were collected by retro-orbital bleeding or sampling from the portal vein. To acquire a PPP, we centrifuged twice to ensure no platelet contamination of supernatant; initially at 2500 rpm for 10 minutes and then 4000 rpm for further 10 minutes. (Page 16, Line 15-16): Human PPP serotonin levels were measured using ClinRep high-performance liquid chromatography kit (Recipe, Munich, Germany) at GreenCross LabCell (Yongin, Korea).

Comment 6) In fig 1b, portal blood 5-HT levels are set at 100%. Why are human portal blood serotonin levels expressed as % when the rodent ones expressed as absolute levels? For consistency, please provide absolute levels of 5-HT in all figures.

We showed absolute levels in Supplementary Fig. 1a and converted this value to % in Supplementary Fig. 1b and Fig. 1b.

Supplementary Fig 1

Fig 1

Comment 7) There is a lot of variation in the group n numbers for the various measures (e.g. 3 or 4 for the 5-HT measurements in figure 1 but 7-10 for the hepatic tricycleride levels). Why are 5-HT levels not provided for all animals?

These are different sets of experiment. Figure 1c, 1d, 1e are results from standard chow diet (SCD) or high fat diet (HFD) fed C57BL6/J mice. However, Figure. 1f, 1g, 1h are results from SCD or HFD fed wild type littermates (*Tph1*^{flox/flox}) and *Tph1* GKO (*Villin-Cre*^{+/-}; *Tph1*^{flox/flox}) mice.

Fig 1

Reviewer #2

The authors have recently shown that short term treatment of TPH inhibitors prevents development of hepatic steatosis in mice fed a high carbohydrate diet. In this manuscript, the authors extend these studies and explore the mechanism and function of 5-HT signals regarding liver energy metabolism *in vivo*. Results show link gut-derived 5-HT, through direct actions on liver, to the pathogenesis of hepatic steatosis. In order to investigate the functional role for gut-derived 5-HT the authors generated gut-specific *Tph1* knockout (KO) mice and induced hepatic steatosis with 8 weeks of HFD diet. Hepatic lipid droplet accumulation, NAFLD activity score (NAS), and hepatic TG levels were dramatically reduced in the liver of HFD-fed *Tph1* GKO mice. In order to identify the relevant metabolic pathways that prevent hepatic steatosis in *Tph1* GKO mice, the authors performed gene expression analyses for an extensive array of metabolic markers. In addition, hepatic *Htr2a* loss-of-function mice were protected against HFD-induced hepatic steatosis. Therefore, the authors evaluated selective HTR2A antagonism as a novel therapeutic strategy for NAFLD. Results presented indicate that selective HTR2A antagonist may prevent hepatic steatosis in HFD-fed mice likely through common mechanisms shared with *Tph1* GKO and *Htr2a* LKO mice. Results highlight the functional importance of hepatic HTR2A signalling in the progression of NAFLD. Inactivation of HTR2A signalling in the liver using *Htr2a* LKO, attenuated hepatic steatosis independent of systemic energy homeostasis.

Fig. 1:

- 1) Serotonin genes are expressed in the small intestine.
- 2) Serotonin levels are higher in portal as compared to peripheral venous blood.
- 3) High-fat diet increases 1) and 2), as expected

4) *Tph1G KO results in improved NAS and hepatic triglyceride levels. The authors should include the data showing (as indicated in the text) that circulating serotonin is reduced. This seems to be an important finding.*

Fig. 2

5) *Tph1G KO does not affect systemic energy homeostasis.*

Fig. 3

6) *Tph1 GKO in general express less genes involved in hepatic lipogenesis. This seems to be expected if HTR2A signalling is reduced.*

Fig. 4

7) *Confirms the role of HTR2A.*

Fig. 5

Sarpogrelate acts as expected as a HTR2A agonist. This compound has been tested vs. aspirin in secondary prevention of cerebral infarction. A published subgroup analysis did not find any differences in individuals without T2DM, dyslipidaemia which indirectly would have supported the role of HTR2A agonists. Verification of the results in humans is therefore necessary.

Comment on Fig. 1. 4) Tph1G KO results in improved NAS and hepatic triglyceride levels. The authors should include the data showing (as indicated in the text) that circulating serotonin is reduced. This seems to be an important finding.

Circulating 5-HT levels in *Tph1* GKO mice was already reported that plasma 5-HT level was about 50% decreased in *Tph1* GKO mice compared to WT littermates (Cell Metab 16, 588-600, 2012).

Comment on Fig. 5) Sarpogrelate acts as expected as a HTR2A agonist. This compound has been tested vs. aspirin in secondary prevention of cerebral infarction. A published subgroup analysis did not find any differences in individuals without T2DM, dyslipidaemia which indirectly would have supported the role of HTR2A agonists. Verification of the results in humans is therefore necessary.

We agree that further verification of our results in human is necessary. The published human data show that sarpogrelate does not affect the metabolic profiles in human subjects. However, those papers did not intend to test the effect of sarpogrelate on fatty liver. Those papers were to test its anti-platelet effects in human. Actually, sarpogrelate or other HTR2A antagonist has never been used to treat metabolic diseases such as diabetes, obesity, and NAFLD. As such, those data need to be carefully interpreted and do not mean that sarpogrelate does not have efficacy in NAFLD. Rather, sarpogrelate is not likely to affect the metabolic profiles in standard condition. We also showed that *Htr2a* LKO and *Tph1* GKO mice are metabolically comparable to wild type littermates when they were fed with standard

chow diet (SCD). These data match well with the published human data. Thus, we need to further confirm the effects of sarpogrelate on human NAFLD, as we have confirmed in mice. In this manuscript, we just provided some human data to suggest the possible importance of our study in human NAFLD. Since human study will take a lot more time and efforts, we decided to share our data with people in the field and motivate to test this model in human.

Comment 1) The data presented seems rather preliminary and it is necessary to further identify detailed molecular mechanisms on how GDS regulates hepatic lipid disposal in the liver and how inhibition of HTR2A signaling reduces hepatic TG accumulation.

We appreciate reviewer's constructive suggestion. Since we did not show detailed downstream molecular mechanism of HTR2A in the liver, our data presented in this manuscript may look a little preliminary. However, we showed the role of gut derived 5-HT on hepatic lipid accumulation using two different tissue specific KO mouse models. We furthermore confirmed this phenotype using pharmacological inhibition of HTR2A. Thus, we have shown comprehensively that inhibition of 5-HT signaling through gut-liver axis can inhibit the initiation/progression of NAFLD. We also provide molecular mechanism showing the changes in hepatic gene expressions. Our gene expression data indicate that de novo lipogenesis and TG synthesis pathways are downregulated in *Htr2a* LKO and *Tph1* GKO mice. In addition, we showed decreased expression of SREBP-1c and PPAR γ which can explain the downregulation of gene expressions of de novo lipogenesis and TG synthesis. Taken together, we already put large amount of data in this manuscript and showed our results comprehensively at organismal and cellular levels. We think that more detailed downstream molecular mechanism of HTR2A is beyond the focus of this manuscript.

Instead, we performed RNA-seq analysis in HFD-fed *Htr2a* LKO mice and WT littermates to provide more comprehensive data on the changes in global gene expressions. The results showed that gene sets which contribute to inflammation and fibrosis in nonalcoholic steatohepatitis (NASH) pathogenesis as well as gene sets that contribute to steatosis were significantly downregulated in *Htr2a* LKO mice.

We added these data and revised RESULTS as follows (Page 8, Line 18 ~ Page 9, Line 4): To further explore the molecular pathways underlying the decreased hepatic lipid accumulation in HFD-fed *Htr2a* LKO mice, we performed RNA-seq and profiled the liver transcriptomes of HFD-fed *Htr2a* LKO mice and WT littermates. We analyzed gene ontology (GO) gene sets by gene set enrichment analysis (GSEA). Among 5917 GO gene sets, 3614 gene sets with sufficient number of matched genes were analyzed and 617 gene sets and 27 gene sets were identified to be significantly enriched in WT littermates and *Htr2a* LKO mice, respectively (Supplementary Table 1). Interestingly, gene sets which contribute to inflammation and fibrosis in nonalcoholic steatohepatitis (NASH) pathogenesis as well as gene

sets that contribute to steatosis were significantly less enriched in *Htr2a* LKO mice (Fig. 5a-c). qRT-PCR analysis further confirmed that proinflammatory and profibrogenic genes were downregulated in the liver of HFD-fed *Htr2a* LKO mice (Fig. 5d). These results suggest that inhibiting 5-HT signaling through hepatic HTR2A can inhibit the progression of NAFLD.

Fig 5

Comment 2) As the authors discuss, there may be more various functions of 5-HT in different tissues. Therefore, a more detailed study of the different roles of 5-HT using tissue specific KO strategy is necessary to further our understanding of the function of the neurotransmitter.

We agree the reviewer's opinion. Since we showed that gut derived 5-HT does not affect systemic energy homeostasis, we need to study more on the roles of 5-HT in other tissues with tissue specific KO strategy. Actually, we have been analyzing the phenotypes of adipocyte-specific *Tph1* KO (*Adiponectin-Cre*^{+/-}; *Tph1*^{fllox/fllox}, *Tph1* FKO) mice. The key phenotypes of *Tph1* FKO mice, decreased body weight gain by increased energy expenditure, are similar with the phenotypes of adipocyte-specific inducible *Tph1* KO (*aP2-CreERT2*^{+/-}; *Tph1*^{fllox/fllox}, *Tph1* AFKO) mice, as we published previously (Nat Commun 6, 6794, 2015). Since adipocyte-derived 5-HT has distinct physiological roles compared to gut derived 5-HT, we are preparing a separate manuscript for these results. As such, we think studies on *Tph1* KO in other tissues will provide different functions of peripheral 5-HT in different tissues and those are beyond the scope of this manuscript.

Comment 3) Do the mice display differences in eating and behaviour? Studies in metabolic cages are necessary.

We share this concern with the reviewer. However, it is already documented that inhibiting peripheral 5-HT synthesis genetically or chemically does not alter the eating behavior (Nat Med 21, 166-172, 2015; Nat Commun 6, 6794, 2015). All of the mice we used in this paper showed similar weight gain upon high fat diet indicating that *Tph1* GKO mice and *Htr2a* LKO mice are not likely to affect eating behavior.

Comment 4) Hepatic lipids should be characterized using lipidomics technology

We thank reviewer's suggestion. We performed lipidomic analysis with the liver of HFD-fed *Htr2a* LKO mice and wild type littermates. *Htr2a* LKO mice exhibited significantly reduced levels of a variety of triglyceride and diglyceride species compared to the wild type littermates (Additional Fig 1 with legend), which is consistent with the liver histology and total triglyceride measurement data (Fig. 4d-f). On the other hand, polar lipid species abundant in membrane structures such as phospholipids and ceramides were similar or higher in the *Htr2a* LKO mice, indicating specific reduction of triglycerides and diglycerides, the hallmark of fatty liver. Thus, the lipidomics data support the notion that liver-specific deletion of *Htr2a* is protective against HFD-induced hepatic steatosis.

Additional Fig 1

Additional Figure 1. (a-b) 12 week old WT and *Htr2a* LKO mice were fed HFD for 8 weeks. **(a)** Orthogonal partial least squares discriminant analysis (OPLS-DA) plots indicate clear separation of WT and *Htr2a* LKO liver lipidome profiles obtained from UPLC/QTOF MS spectra for positive and negative modes. **(b)** Relative abundance of triglyceride (TG), diglyceride (DG), phosphatidylethanolamine (PE), phosphatidylcholine (PC), phosphatidylglycerol (PG), and ceramide (Cer) species in the liver tissues. Only significantly different (VIP score of >2 in the OPLS-DA model and P -value <0.05) species are shown. $n=4$ per group. Data are expressed as the means \pm standard error of the mean (SEM). * P <0.5, ** P <0.01, *** P <0.001, Student's t -test (**b**).

Comment 5) Suppl figure 1 c; significance seems to be driven by one single data point

In order to overcome this limitation, we continued to enroll patients in this study. After initial submission of our manuscript, we further analyzed 3 more patients (total 9 patients) and the tendency of the results was similar with the results we described initially.

We revised RESULTS as follows (Page 5, Line 10-13): Also, the ratio of 5-HT between portal and peripheral blood showed a tendency of positive correlation with markers associated with NAFLD such as blood levels of alanine transferase (ALT), gamma-glutamyltransferase (GTP), triglycerides (TG), and transient elastography controlled attenuation parameters (CAP) (Supplementary Fig. 1c).

Reviewer #3

This paper by Choi et al presents studies identifying a role for gut-derived serotonin synthesis in de novo lipogenesis and hepatic steatosis via reduction in liver serotonin receptor 2A (HTR2A) signalling. They showed increased serotonin concentrations in the portal blood of six liver living donor subjects as well as in mice on a high-fat diet. Both gut-specific Tph1 knockout mice and liver-specific Htr2a knockout mice were resistant to short-term HFD-induced hepatic steatosis without having any effects on systemic lipid, glucose, and energy metabolism. Finally, the use of a selective HTR2A antagonist treatment mimicked the protective effects found on the KO on HFD-induced hepatic steatosis. They concluded that gut TPH1-liver HTR2A axis is a promising novel drug target for NAFLD. The manuscript is interesting and of potential significance. There are number of concerns with the current dataset.

Major issue 1) The human data presented in figures 1 is very hard to interpret. These are living donors were the presence of steatosis/NAFLD is strictly ruled out by extensive work up including liver biopsy that portal blood was obtained from at the time of transplantation. The authors showed correlation with

serotonin levels in portal blood with various markers of steatosis and NAFLD progression??

We understand this concern. What we want to show in Fig. 1. is the positive correlation of the portal/peripheral blood 5-HT ratio with the clinical indicators that are known to be associated with NAFLD. It would have been better if we checked these in NAFLD patients. But as you know, there are many limitations in clinical study and thus it is practically quite difficult to collect portal blood in NAFLD patients. In this manuscript, we take advantage of human data to show the relevance of our concept that gut derived 5-HT can affect liver through portal vein.

We revised RESULTS as follows (Page 5, Line 10-13): Also, the ratio of 5-HT between portal and peripheral blood showed a tendency of positive correlation with markers associated with NAFLD such as blood levels of alanine transferase (ALT), gamma-glutamyltransferase (GTP), triglycerides (TG), and transient elastography controlled attenuation parameters (CAP) (Supplementary Fig. 1c)

Major issue 2) In both humans with NAFLD and mice on HFD the main mechanism of steatosis is increased uptake of free fatty acids from circulation mainly via increased lipolysis in the context of systemic and adipose tissue insulin resistance. The current study showed that gut-specific Tph1 knockout mice and liver-specific Htr2a knockout mice on a short-term HFD for eight weeks have no effects on systemic lipid, glucose, and energy metabolism. These results strongly suggest that the effects on steatosis noted are likely to be transient and disappeared with more prolonged feeding. In fact, these diets are typically given 16 to 24 weeks in NAFLD related research. Thus at least a later time point (16 or 24 wks) should be included to better understand the impact of GDS on NAFLD.

We agree that the effects on the steatosis were transient and disappeared with more prolonged HFD feeding. We have tested longer HFD feeding for 32 weeks from 12 weeks of age to 44 weeks of age and hepatic TG levels of *Htr2a* LKO mice

were similar with WT mice (Additional Fig. 2 with legend). Since hepatic TG accumulation is determined by the balance between calorie input and output in the liver, any anti-steatosis treatment is transient and prolonged increase in calorie input ultimately induces steatosis, unless calorie input is completely blocked. However, considering the broad disease spectrum of NAFLD, slowing the development or progression of hepatic steatosis is an effective strategy for treating NAFLD. Likewise, inhibition of 5-HT signaling through gut-liver axis can be effective in treating NAFLD.

Additional Fig 2

Additional Figure 2. (a-f) 12 week old WT and *Htr2a* LKO mice were fed SCD or HFD for 32 weeks. (a) Representative liver histology by H&E staining from HFD fed WT and *Htr2a* LKO mice. Scale bars, 100µm. (b) NAS of HFD fed WT and *Htr2a* LKO mice. n = 5-8 per group. (c) Hepatic triglyceride levels. n = 5-8 per group. (d) Body weight trends. n = 5-8 per group. (e) Intraperitoneal glucose tolerance test (IPGTT) after 16hr fasting. n = 5-8 per group. (f) Organ weight to body weight ratio. n = 5-8 per group.

We also agree that 8-week HFD feeding is relatively short. The longer HFD feeding (16~24 weeks) is commonly used to study NAFLD in mouse model because the key features of NAFLD, increased fatty acid uptake and de novo lipogenesis (DNL), are more robustly observed when mice are fed with HFD for longer periods of time. C57BL6/J mice usually show better response to HFD feeding compared to other mice strains and 8-week HFD feeding is enough to induce insulin resistance and glucose intolerance, which increases hepatic fatty acid uptake and DNL in C57BL6/J mice. In this manuscript, we maintained all the mice in pure C57BL6/J background. Transgenic mice were backcrossed with C57BL6/J mice more than 10 generations to achieve pure C57BL6/J background before we included them in experimental cohort. Thus, we could clearly show the development of NAFLD phenotypes in many ways after 8-week HFD feeding. We also observed the typical changes in hepatic gene expressions during progression of NAFLD after 8-week HFD feeding. In this context, NAFLD development was prevented and the

expression of genes involved in DNL was downregulated in both *Tph1* GKO and *Htr2a* LKO mice. Furthermore, we performed RNA-seq analysis in 8-week HFD-fed *Htr2a* LKO mice, which provide more comprehensive data on the changes in global gene expressions. The RNA-seq results showed that gene sets which contribute to inflammation and fibrosis in nonalcoholic steatohepatitis (NASH) pathogenesis as well as gene sets that contribute to steatosis were significantly less enriched in *Htr2a* LKO mice. These data indicate that 8-week HFD is enough to show NAFLD phenotype in our experimental setting.

Major issue 3) The suggested effects on the novo lipogenesis in the liver are not fully supported with only limited data on gene expression of various enzymes involved in lipid metabolism provided.

We agree that our gene expression studies only partly support the change in hepatic de novo lipogenesis (DNL). Measuring DNL in vivo is the best and the most straightforward method to test the effects of 5-HT on the hepatic DNL. However, measuring DNL in vivo is technically difficult and we are currently not able to do this experiment. Instead, we checked hepatic gene expressions thoroughly in several different mouse models. Genes involved in DNL are well known to be regulated at transcription level and SREBP-1c and PPAR γ are known as master regulator of DNL genes in the liver. Thus, transcriptional changes of these genes strongly indicate the change in hepatic DNL. In the present manuscript, we showed that HFD increased the expression of SREBP-1c and PPAR γ as well as genes involved in DNL and inhibition of 5-HT signaling downregulated those gene expressions, which strongly suggested that 5-HT signaling through HTR2A in the liver is necessary to regulate hepatic DNL upon HFD feeding.

Major issue 4) The studies using sarpogrelate as a therapeutic strategy for NAFLD does not take into account the effects on platelet aggregation and on soluble adhesion molecules induce by this drug that may have a significant impact on the liver phenotype observed. Studies using RNA-based therapy to selectively disrupt the GDS would be warranted.

We share reviewer's concern. Sarpogrelate is known to inhibit platelet aggregation and decrease soluble adhesion molecules. However, we don't think that the effects on platelet aggregation and soluble adhesion molecules have a major impact on the hepatic phenotype of HFD-fed mice with sarpogrelate administration. Before we tested sarpogrelate, we showed the role of HTR2A in the development of fatty liver using *Tph1* GKO and *Htr2a* LKO mice. Data from these genetic models can exclude the possibility of off-target effects. In addition, we have tested liver-specific *Htr2b* KO (*Albumin-Cre*^{+/-}; *Htr2b*^{flox/flox}, *Htr2b* LKO) mice to exclude the potential effect of HTR2B. HFD-fed *Htr2b* LKO mice showed no difference in

histology, NAS, and hepatic TG levels compared to WT littermates. We added these data in our revised manuscript.

We added these data in our revised manuscript as follows (Page 8, Line 10-17): As HTR2B is the most highly expressed hepatic HTR and has been associated with hepatic carbohydrate metabolism, we generated liver-specific *Htr2b* KO (*Albumin-Cre*^{+/-}; *Htr2b*^{flox/flox}, *Htr2b* LKO) mice to examine if inactivating hepatic HTR2B signaling pathway has anti-steatotic effects in the liver. HFD-fed *Htr2b* LKO mice showed no difference in liver histology, NAS, and hepatic TG levels compared with WT littermates (*Htr2b*^{flox/flox}) (Supplementary Fig. 3a, b, c). From these results, we confirmed that the protective effect against hepatic steatosis by reducing GDS production was highly specific for hepatic HTR2A.

Supplementary Fig 3

REVIEWERS' COMMENTS:

Reviewer #1 (Remarks to the Author):

I have no further comments or queries for the authors - they have adequately addressed my earlier concerns

Reviewer #2 (Remarks to the Author):

The authors have not adequately addressed the comments; they state "since human study will take a lot more time and efforts, we decided to share our data with people in the field and motivate to test this model in human" and "since adipocyte-derived 5-HT has distinct physiological roles compared to gut derived 5-HT, we are preparing a separate manuscript for these results. As such, we think studies on Tph1 KO in other tissues will provide different functions of peripheral 5-HT in different tissues and those are beyond the scope of this manuscript".

This reviewer respects this decision, but it reduces the interest in this study.

Reviewer #4 (Remarks to the Author):

The authors have performed a meaningful revision of manuscript making several improvements. I have no additional comments.